# Statistical and Visual Analysis of Audio, Text, and Image Features for Multi-Modal Music Genre Recognition

**DOI:** 10.3390/e23111502

**Published:** 2021-11-12

**Authors:** Ben Wilkes, Igor Vatolkin, Heinrich Müller

**Affiliations:** Department of Computer Science, Technische Universität Dortmund, 44227 Dortmund, Germany; ben.wilkes@tu-dortmund.de (B.W.); heinrich.mueller@tu-dortmund.de (H.M.)

**Keywords:** music genre recognition, multi-modal classification, feature evaluation, audio signal features, album cover images, lyrics

## Abstract

We present a multi-modal genre recognition framework that considers the modalities audio, text, and image by features extracted from audio signals, album cover images, and lyrics of music tracks. In contrast to pure learning of features by a neural network as done in the related work, handcrafted features designed for a respective modality are also integrated, allowing for higher interpretability of created models and further theoretical analysis of the impact of individual features on genre prediction. Genre recognition is performed by binary classification of a music track with respect to each genre based on combinations of elementary features. For feature combination a two-level technique is used, which combines aggregation into fixed-length feature vectors with confidence-based fusion of classification results. Extensive experiments have been conducted for three classifier models (Naïve Bayes, Support Vector Machine, and Random Forest) and numerous feature combinations. The results are presented visually, with data reduction for improved perceptibility achieved by multi-objective analysis and restriction to non-dominated data. Feature- and classifier-related hypotheses are formulated based on the data, and their statistical significance is formally analyzed. The statistical analysis shows that the combination of two modalities almost always leads to a significant increase of performance and the combination of three modalities in several cases.

## 1. Introduction

Music genre recognition is one of the most common classification tasks in music information retrieval, with several hundreds of published studies mentioned by Sturm [1]. Traditional approaches are usually based on an individual feature source, mainly the audio signal. Because different modalities beyond audio, such as text, images, or symbolic representations, may contain complementary information, multi-modal approaches bear great opportunities to improve the classification performance. In this work, we present a multi-modal genre recognition framework that considers audio, text and image features of a music track by features of audio tracks, album cover images, and lyrics.

Because, in the field of image and text classification, artificial neural networks achieved comparatively good classification results to date [2,3], a group of text- and image-based features computed by artificial neural networks is taken into account in our framework. However, the features automatically learned by a neural network are often less interpretable and can also have a poor generalization ability because of a typically very large number of parameters of a trained neural network. Therefore, a further group of handcrafted text- and image-based features is additionally employed, which have been successfully used for image or text classification tasks in the past. For audio, several groups of features related to harmony, rhythm and tempo, timbre, and musically meaningful semantic properties from previous work predicted by supervised classification models are considered.

Genre recognition is performed based on binary classification of a music track with respect to each genre. From the results of the genre classifiers, the membership to one of the genres is predicted, and a confidence value for this prediction is given. Besides combination of features into fixed-length feature vectors, a second approach of feature combination in form of confidence-based fusion of predictions obtained from several feature vector-based predictions is employed. This allows a detailed representation of longer audio tracks by a length-dependent number of feature values. Combinations of text features and combinations of image features, as well as combinations of text and image features are handled by feature vectors, whereas combinations of audio features and of audio, text, and image features are handled by confidence-based fusion.

A special focus is placed on performance analysis. To assess the influence of classifiers and features on the quality of genre recognition, we have determined the balanced classification error experimentally for three classifiers and numerous feature combinations. The resulting error values are presented visually. To improve perceptibility, data reduction techniques based on multi-objective analysis and restriction to non-dominated data are proposed and applied. Based on these data, feature- and classifier-related hypotheses are formulated and their significance is statistically tested. The global finding is that the combination of features from two modalities yields a significant reduction of the classification error for the majority of use cases. The extension to three modalities leads in several cases to further significant improvements.

The following Section 2, presents a review of related work and the contributions of the paper in this context. Section 3 describes our approach in detail. Section 4 deals with the experimental evaluation of the proposed system. Section 5 provides a summary of results and an outlook on possible future work.

## 2. Related Work

Common approaches to genre classification focus on audio features and their combinations. This is motivated by the situation that audio features describe or correlate to many different musically meaningful properties of a music piece and can be extracted when the digital score is not available. In one of the first related studies, Tzanetakis and Cook [4] introduced features to represent pitch, rhythmic structure, and timbre. Based on these features and their combinations, a Gaussian classifier for music genre classification was trained. Lidy and Rauber [5] presented different rhythm characteristics and compared their performance when used for music genre classification. In addition, the influence of psychoacoustic transformations on rhythm features was considered to improve the classification performance. Scaringella et al. [6] also provided an overview of various audio features describing timbre, harmony, and rhythm, and examined the impact of using different classifiers.

One possibility for applying image classification methods for genre classification is the use of direct visual representation of music. Bainbridge and Bell [7] and Burgoyne et al. [8] extract musical notes and lyrics from images of scores. Another concept is to convert the audio signal into a two-dimensional image representation (e.g., a spectrogram) and to apply further image processing methods. For instance, Ke et al. [9] used spectrograms to identify related music pieces. Another option is to use image-based information, which is often associated with music, especially album covers, photographs, and videos. Dorochowicz and Kostek [10] conducted a study with the aim to find out whether there exists a relationship between typographic, compositional, and coloristic elements of the music album cover design and genre of the music contained in the album. Le [11] measured the color similarities of album covers based on various genres and presents a study to verify whether the average listener can determine the genre of contemporary albums based on the graphics displayed on album covers. Schindler [12] discussed the role of visual information for music information retrieval and music genre classification, presents methods for the use of image information, analyzes them on the basis of images from music videos, and draws conclusions about their significance for album covers, as well. Oramas et al. [2,3] used album covers as image component for multi-modal genre classification from audio, text, and images. This work will be discussed in more detail later in this section. Libeks and Turnbull [13] presented an image classification system that is able to estimate the similarity of music artists or to determine related genres based on album covers and photos of the artists. A data set of artists was built along with genre annotations and their most popular album covers and photos. The classification system calculates, for each photo and cover of a given artist, the most similar image from the data set. For each resulting image, the genre annotations of the associated artist were collected and then averaged over the data set, leading to genre prediction for a given artist.

Lyrics are more commonly used as an information source than album cover art. Logan et al. [14] estimated the similarity of artists based on their lyrics and compared the results with an audio-based approach, which achieved better results. The authors suggested to combine audio and text features to get better results. Other studies applied lyrics features for mood prediction [15,16].

The combinations of features from different sources for music classification are until now not very thoroughly explored. In the following, we provide some references. For a recent overview, we refer to Simonetta et al. [17].

Most studies on multi-modal music classification combine two sources. Audio, together with lyrics, seems to be the most frequent case. These sources were applied for genre recognition [18,19,20,21], mood and emotion recognition [22,23,24,25], artist identification [26], hit song prediction [27], and playlist prediction [28]. Audio and symbolic features were used for genre recognition [29,30]. Audio and images were employed for mood prediction [31] and genre recognition [12].

Rather few studies addressed three and more feature sources. Audio, cultural, lyrics, and symbolic descriptors were combined for genre recognition by McKay et al. [32] and audio, symbolic, and lyrics descriptors for mood detection by Panda et al. [33].

To our knowledge, the papers by Oramas et al. [2,3] are the only published works that deal with music genre classification on the basis of image, text, and audio-based features. Three separate artificial neural networks were trained on album covers, audio tracks, and album reviews. As inputs, the audio signals were converted into spectrograms and the album reviews into a bag-of-words representation. After the training, the three resulting networks were combined into a new network by reconnecting some layers and re-training. This network is used in our work for the extraction of image features.

Although a general concept of our framework is inspired by Oramas et al. [3], there exist several important differences. First, we also take handcrafted features into account, but apply classification methods for genre prediction, which have significantly fewer parameters than deep neural networks. This can help to create more interpretable models based on semantic features and has a further advantage in that the models can be trained with very small data sets. For example, when a listener defines a new category based on only a few representative tracks, models with many parameters, such as neural networks, will tend to overfit in that real-world application scenario. Second, we estimate text features only from lyrics and not album reviews. Although it is argued by Oramas et al. [3] that relevant genre information must not be captured in reviews, and, thus, reviews “will unlikely comply with the current taxonomy of the collection to be classified”, it is a safer way to consider only lyrics. Third, audio tracks in reference [3] are always represented with 15-s frames, as the convolutional networks expect a fixed-size input. However, particularly for more complex genres and styles with very different segments, the analysis of complete music tracks may be useful and important information may be omitted when a frame of a fixed size is used for each track independently of its length. We handle this issue by considering multiple fixed-length frames by confidence-based feature combination.

Further differences between our work and [3] include (a) the approach to fuse the results of classification models for each modality based on the confidence level, which is estimated differently for individual modalities (Section 3.5), (b) the method for visualizing experimentally collected performance data for comparing the influence of different feature combinations and classifier models (Section 4.2), and (c) rigorous statistical testing of hypotheses, which underlines that, in some cases, the combination of several modalities does not necessarily lead to a significant improvement of the classification quality (Section 4.3).

## 3. A Multi-Modal Approach to Music Genre Recognition

In the following, we present the backgrounds and the details of our framework. Section 3.1 starts with a brief discussion on music genres and a description of our data set. Section 3.2 provides an overview of our approach. Section 3.3 describes audio-, text-, and image-based features used in our study. Section 3.4 briefly summarizes the classification algorithms used. The fusion of classification models trained separately for individual modalities is introduced in Section 3.5.

### 3.1. Data Set

Moore [34] refers to a *music genre* as a set of musical events, the scope of which is determined by specific generally accepted rules. Often, music pieces of the same genre have similar characteristics in instrumentation, rhythm structure, and pitch content [4]. Music genres, however, have never been formally defined [35], so that the assignment of music pieces to genres is often a matter of personal interpretation. In particular, music pieces could be assigned to different genres at the same time.

Therefore, the genre annotations used in our work are subjective and represent only one possible scenario. Because not all modalities can always be automatically extracted, we have created an in-house multi-modal data set of 446 tracks compiled from several music collections: 1000 songs, 1517-artists, SALAMI, SLAC, and an album collection of TU Dortmund. Appendix A provides details about these collections. The genres to predict are *Rock Rap/Hip-Hop*, *Electronic*, *Folk/World/Country*, *Blues*, *R&B*, *Jazz*, *Pop*, *Classical*, and *Reggae* (sorted by the number of corresponding tracks). Each music track is assigned to exactly one genre. Appendix B provides the details of the reassignment of the music genres of the original data sets to the newly created data set. Figure 1 shows the distribution of the genres.

To find album covers, music texts, and genres, the Internet databases of Discogs [36] and MusicBrainz [37] for album covers, the Internet databases of MetroLyrics [38], LyricWiki [39], CajunLyrics [40], Lololyrics [41], and Apiseeds Lyrics [42] for lyrics and the database of Discogs for genres were used, in that order.

### 3.2. General Approach

We treat music genre recognition as a classification problem, which maps objects (here, music tracks) to classes (here, genres). We adopt the two-step approach of classification, which first assigns features to the objects and then uses them to perform the classification. Parametrized statistical models are employed as classifiers, which are trained in a preprocessing step by supervised learning. The training procedure adjusts the parameter values, so that the objects of a given training set, whose classes are known, are classified as correctly as possible.

We use two types of features. The first type are features that have proven to be particularly useful in the field of audio, text, and image classification. The second type of features result from classifying artificial neural networks. Such neural networks combine feature assignment and classification. The features depend on parameters, whose values are determined simultaneously with the parameters of the classification step by training. We use features computed in this way analogous to the features of the first type. Section 3.3 provides an overview of the features employed in this paper.

Genre recognition is performed by binary classification of a music track with respect to each genre based on combinations of elementary features. A binary classifier is assigned to each genre, which decides whether a music track belongs to the genre (a positive prediction) or does not belong to the genre (a negative prediction). We employ three classifier models, Naïve Bayes, Support Vector Machine, and Random Forest, which are briefly recalled in Section 3.4.

For feature combination a two-level technique is used. The first level is feature aggregation into fixed-length feature vectors. Combinations of text features and combinations of image feature, as well as combinations of text and image features are handled in this way. From the results of applying all genre classifiers to such a feature vector of a piece of music, the membership to one of the genres is predicted, and a confidence value for this prediction is given. The second level of feature combination is confidence-based fusion of predictions obtained from several feature vector-based predictions. Combinations of audio features and of audio, text, and image features are handled in this way. Section 3.5 presents the details of this approach.

### 3.3. Features

In the following, we present audio, text, and image features estimated from audio tracks, lyrics, and album covers.

#### 3.3.1. Audio Features

Audio features are calculated from 22,050 Hz mono wave files converted from original mp3 tracks. The description and grouping of the audio features described below is based on previous work [43].

##### Tempo and Rhythm

A typical characteristic of the temporal progress of a music piece is the number of beats per minute, where the beat events correspond to perceived sound pulses with highly repetitive structure. The rhythm is described by the special arrangement of the note lengths and accentuation in a music piece [44]. To describe the rhythm of a music piece, for example, the change in the loudness of certain sub-frequency bands can be examined, such as fluctuation patterns [45]. Rhythm must be differentiated from tempo because a particular rhythm pattern can be played in different tempo; therefore, they are not firmly connected. However, rhythm and tempo are strongly related, as they describe the temporal aspect of the music piece. They often consist of autocorrelation (the correlation of the audio signal with itself after an additional time lag). Section C.1 lists all tempo and rhythm features, together with their dimensionality and related references.

##### Timbre

Timbre can be defined as the part of the auditory sensation that allows the listener to distinguish between two sounds that have the same loudness and pitch [46]. The timbre depends, for instance, on the instrument used or the way it is played. Features that describe the timbre can be grouped by their extraction domains, such as time domain (e.g., the root mean square energy), spectrum (spectral centroid), cepstrum (MFCCs), or phase domain (angles in the phase domain). Section C.2 lists all timbre features used in our study.

##### Harmony

Harmony can be defined as the relationship between simultaneously played notes and the way how these relationships change over time, cf. reference [47]. The difference in tone frequencies between two notes played at the same time is called an “interval”. Intervals may be consonant or dissonant, i.e., sounding pleasing/perfect or unpleasing/tense to listeners, which, however, can be perceived subjectively. The ratio of consonant to dissonant intervals is central to the study of the harmony of music. In addition, the transform of the frequency amplitudes to the halftones (chromagram or pitch class profile) can be treated as a harmonic feature because it serves as a base feature for more complex properties, such as chords or keys. Section C.3 shows the harmony features.

##### Semantic Features

Semantic features describe characteristics of the piece of music, which are related to music theory, such as the instrumentation, characteristics of the voices in the song, or the mood expressed. To capture semantic features from digitally represented music, various classifiers have been trained on a set of audio features, using multi-objective feature selection and ensembles of classifiers, with some semantic features derived or also predicted from other semantic features as introduced in reference [43]. The corresponding descriptors are listed in Section C.4.

#### 3.3.2. Text Features

Two text feature groups are used, which are induced by the multidimensional Bag-of-Words feature and the doc2vec feature described below.

##### Bag-of-Words (BoW) Feature

In its simplest version, the BoW text feature [48] measures the occurrence frequencies of words from a given domain of words. The result is a real vector whose components correspond to the words of the domain. Before the feature estimation some preprocessing procedures are typically applied [49] (p. 242). In this work, stop words, such as “is”, “to”, or “with”, are removed, and words are substituted with their stems, such as “lov” for “lover” and “loving”. Furthermore, the frequency of a word is measured with the *Term Frequency-Inverse Document Frequency* (TF-IDF). TF-IDF is the product of the relative frequency and the *Inverse Document Frequency* (IDF). IDF is the inverse of the relative frequency of occurrence of a word in a document from the document collection under consideration. The motivation for using TF-IDF is that infrequent words could describe important text properties.

Advantages of BoW features are the fast estimation and high comprehensibility. Disadvantages are the loss of information about the order of the words, as well as a possible high dimension of the feature vectors, which depends on the number of different words used.

##### Doc2vec Feature

As in the BoW feature, the doc2vec feature [50] comes from the discipline of document classification. The doc2vec feature extends the idea of the word2vec-feature [51].

Word2vec and doc2vec are methods for the numerical representation of words or documents in a vector space. One simple option for such a representation is *one-hot-coding*. This means that, for every possible word or document, respectively, there is exactly one vector component whose value is 1 for the represented word, and 0 otherwise. In contrast to this, word2vec methods represent the words of a vocabulary in a latent space, which has a lower dimension than one-hot-encoding and aims to store context information of words. Doc2vec expands the latent space by a representation of documents by low-dimensional vectors, which store the context information document-specifically. In both cases, the vectors result from weights of neural networks.

One approach of word2vec uses fully connected neural networks with one hidden layer that outputs for two input words the probability of all words to occur as the middle word in the context of the input words. The hidden layer calculates a feature vector, which is used by the output layer to determine the context probabilities. The input layer, such as the output layer, has a neuron for each word of the vocabulary, for one-hot coding. Each input neuron is connected to all hidden layer neurons. The weights on these connections form the representing vector of the word. Its dimension and, thus, the dimension of latent space is the number of neurons in the hidden layer. A simple introduction is given by Skansi [52] (Chap. 9).

For doc2vec, input neurons are added for the documents, whose vectors are then constructed accordingly.

Compared to BoW features, doc2vec features have the advantage that the information about the contextual relation of the words is included in the feature calculation. While the number of lyrics increases, the dimensionality of the feature vector does not increase, as for BoW features, because it is not dependent on the diversity of the words in the lyrics, but is an adjustable parameter. By using artificial neural networks, however, the interpretability and the explainability suffer because the semantics of doc2vec features is hardly comprehensible for humans.

#### 3.3.3. Image Features

Two image features groups are used, which are induced by the Bag-of-Features with SIFT descriptors and deep convolutional neural network features.

##### Bag-of-Features (BoF) with SIFT Descriptors

SIFT stands for “*Scale -Invariant Feature Transform*” [53]. SIFT features are local image pixel descriptors and are invariant against rotation, scaling, and displacement. A SIFT descriptor is a 128-dimensional vector, which encodes properties of a pixel and its local environment in the image: a size, a position, an orientation, and further characteristics of its environment.

The BoF feature is an extension of the principle of the BoW feature to other data types than text. In the case of images, a *visual vocabulary* of so-called *visual words* is constructed. A BoF feature produces a real-valued vector, which measures frequencies of each visual word. Frequency measures can be, such as for BoW features, the absolute and relative frequency or TF-IDF. The dimension of the BoF feature depends on the size of the visual vocabulary.

To construct the visual vocabulary, local image features are determined for the images of the training set at first. The resulting set of image features is assigned to *k* clusters by applying a clustering algorithm, where *k* is the desired size of the visual vocabulary. In this work, we apply the *k-means algorithm* by Lloyd [54], which also estimates a cluster center for each cluster. The set of cluster centers forms the visual alphabet.

Being related to the BoW feature, the BoF feature has similar weaknesses. It also loses contextual information because information about the locations of the local image features on the images is neglected. In contrast to BoW features, the size of the vocabulary is a freely adjustable parameter. Nevertheless, it can be assumed that, as the number of images increases, so also does the diversity of descriptors and more cluster centers should be used.

##### Features of Deep Convolutional Neural Networks

*Deep convolutional neural networks* have a high number of hidden layers and are particularly successful for image classification. For this purpose, various network architectures are known, e.g., the ResNet [55]. To estimate a feature vector for the given image, it is used as input in an image classification network. The output values of the last hidden layer, which are also the input values of the classifier section, build the feature vector. In this work, the network by Oramas et al. [2] based on *ResNet101* by He et al. [55] is used to classify the album covers. The obtained features are in the following called “*DNN features*” or “*DNNF*”.

#### 3.3.4. Reduction of Dimension by Principal Component Analysis

The text and image features presented can be high-dimensional, depending on their extraction parameters. This may cause the problem that the number of available pieces of music (see Section 3.1) is not sufficient to train the classifiers with acceptable general performance. For this reason, *Principal Component Analysis* (PCA) [56] is additionally applied to reduce the dimensionality of the corresponding feature vectors.

### 3.4. Classifiers

The supervised classification algorithms listed below operate on numeric feature vectors of fixed dimension.

#### Naïve Bayes

The Naïve Bayes classifier was originally designed by Maron [57] for the classification of text documents. According to Qiang [58], it is very efficient and provides good results in many applications. However, if the structure of the feature vectors deviates strongly from the assumption of independence, the classification quality suffers. It can be assumed that some of the features in our study are dependent, so that this circumstance is a weak point of the classifier in the application of this work.

#### Linear Support Vector Machine

The linear *Support Vector Machine* (SVM) [59] classifies items by placing hyperplanes in feature space and determining the class membership of an item to be classified by the location of its feature vector relative to the hyperplane. The location of the hyperplane is determined by training the SVM. The details are described by Cristianini and Shawe-Taylor [60]. Linear SVMs are known to provide good classification results even for high-dimensional feature vectors and comparatively little training data. Linear SVMs can additionally be calculated very efficiently. However, if the separation of the data of the problem by its position in the feature space cannot be approximated by a linear hyperplane, then, linear SVMs show high error rates.

#### Random Forest

The Random Forest classifier [61] is based on a set of decision trees that vote by majority over the class of a feature vector. The Random Forest uses the *Classification and Regression Trees* (CARTs) [62]. CARTs have many applications in machine learning because they are invariant in scaling and many other transforms of feature vectors. Furthermore, they are also robust against inserting irrelevant data and create models that can be read and understood by humans. However, their classification performances are seldom good (reference [63], p. 352), as they tend to overfit [64]. To counteract this property, the Random Forest classifier uses modified CARTs along with the bagging developed by Breiman [65]. For further information, we refer to the remarks of Au [64].

### 3.5. Fusion of Binary Models Trained for Individual Genres and Modalities

The multi-modal genre recognition in this study is based on binary decisions. This means that, for each genre $g\in \left\{1,\ldots ,G\right\}$, an individually trained classification model indicates whether a given music piece belongs to this genre.

For the fusion of binary models, which predict genres based on individual modalities, we distinguish between three cases to estimate confidences for genre predictions: (1) *audio features only*, (2) *a combination of text and image features only*, and (3) *a combination of audio, text, and image features*. The final decision based on all modalities takes into account the confidences of predictions of the cases (1) and (2), as described below.

In the *subcase* (1), genre predictions are first done on time intervals (classification frames) of 4 s length with 2 s overlap. The aggregation of features along the complete music track would decrease the classification performance because, even for tracks of the same genre, each music piece typically contains several different segments with respect to instrumentation, harmonic, and rhythmic properties. Let $W_{m}$ be the number of classification frames in the music track *m*, which is represented with feature vectors ${\vec{x}}_{1}\left(m\right),\ldots ,{\vec{x}}_{W_{m}}\left(m\right)$. Let ${\hat{y}}_{w}\left(m,g\right)\in \left\{0,1\right\}$ be the prediction for the *w*-th classification frame (equal to 1 when this frame is predicted to belong to the genre *g* and 0 otherwise). The assignment of complete tracks to genres is done by majority voting (index “a” stands for audio):(1)$${\hat{y}}_{a}\left(m,g\right):={\hat{y}}_{a}\left({\vec{x}}_{1}\left(m\right),\ldots ,{\vec{x}}_{W_{m}}\left(m\right),g\right)=\left\lceil \frac{1}{W_{m}}\cdot \sum_{w=1}^{W_{m}}{\hat{y}}_{w}\left(m,g\right)-\frac{1}{2}\right\rceil ,$$
and the *confidence of the prediction* based on audio features is given as:(2)$$c_{a(m,g)}=\left\{\begin{array}{cc} \frac{1}{W_{m}}\cdot \sum_{w=1}^{W_{m}}{\hat{y}}_{w}\left(m,g\right) & \mathrm{if}\;{\hat{y}}_{a}\left(m,g\right)=1\\ 1-\frac{1}{W_{m}}\cdot \sum_{w=1}^{W_{m}}{\hat{y}}_{w}\left(m,g\right) & \mathrm{otherwise}. \end{array}\right.$$

In the *subcase* (2), vectors of text and image features can be simply concatenated because they have the same length for all music pieces. The confidence of the prediction ${\hat{y}}_{\mathrm{it}}\left(m,g\right)$ (index “it” stands for image and text) for music piece *m* and genre *g* depends on the number of positive predictions for all other genres $i\in \left\{1,\ldots ,G\right\}\backslash \left\{g\right\}$:(3)$$c_{\mathrm{it}}\left(m,g\right)=\left\{\begin{array}{cc} 1-\frac{1}{G-1}\cdot \sum_{i\in \left\{1,\ldots ,G\right\}\backslash \left\{g\right\}}{\hat{y}}_{\mathrm{it}}\left(m,i\right) & \mathrm{if}\;{\hat{y}}_{\mathrm{it}}\left(m,g\right)=1\\ \frac{1}{G-1}\cdot \sum_{i\in \left\{1,\ldots ,G\right\}\backslash \left\{g\right\}}{\hat{y}}_{\mathrm{it}}\left(m,i\right) & \mathrm{otherwise}. \end{array}\right.$$
Thus, the highest possible confidence $c_{\mathrm{it}}\left(m,g\right)=1$ is given only if the music piece *m* is assigned to genre *g* by the binary classification model, which predicts this genre and is assigned as not belonging to all other genres by the related classification models.

In the *subcase* (3), the final prediction is made with respect to predictions and confidences of decisions done in *subcases* (1) and (2):(4)$${\hat{y}}_{\mathrm{ait}}\left(m,g\right)=\left\lceil \frac{1}{2}\cdot \left({\hat{y}}_{a}\left(m,g\right)\cdot c_{a}\left(m,g\right)+{\hat{y}}_{\mathrm{it}}\left(m,g\right)\cdot c_{\mathrm{it}}\left(m,g\right)\right)-\frac{1}{2}\right\rceil .$$

Training sets for each classifier are balanced, i.e., they contain the same number of positive (belonging to the genre to predict) music tracks and negative (not belonging to this genre) tracks, in order to avoid a bias of one of the classes. For this purpose, the set of initially available tracks for training is selected as follows. Let V(g) be the number of tracks available, which belong to the genre *g*, and $\bar{V}\left(g\right)$ the number of tracks not belonging to this genre. For the data set described in Section 3.1, $V\left(g\right)<\bar{V}\left(g\right)$ holds for all genres. The number of negative training tracks is reduced to approximately $\bar{V}\left(g\right)$ by first sorting those songs according to their genres and then retaining only every $\left\lfloor \bar{V}\left(g\right)/V\left(g\right)\right\rfloor$-th element.

## 4. Evaluation

The main goal of the evaluation is to understand the influence of modalities and feature groups on music genre recognition. In addition, insights into the performance of different tested classifiers (Naïve Bayes, SVM, and Random Forest), in absolute terms and in comparison, should be gained. For this purpose, we formulate several hypotheses. *Feature-related hypotheses* are addressed in Section 4.3 and *classifier-related hypotheses* in Section 4.4. The focus of the evaluation of feature-related hypotheses is on statements on the effect of feature combinations. Their significance is assessed by statistical tests. Statements on the classifier-related hypotheses are based on the visual analysis of the data, which is discussed in Section 4.2. The configuration of experiments is provided in Section 4.1.

### 4.1. Configuration Of Experiments

The configurations of the text BoW and doc2vec features are summarized in Table 1 and Table 2.

The configurations of the image BoF SIFT and DNN features are provided in Table 3 and Table 4.

The parameters for text and image features were determined experimentally in random samples. For this purpose, a grid search was executed on a strongly reduced version of the training data set. We studied vocabulary sizes of 25, 50, 100, 200, and 400 for doc2vec, BoF, and BoW features, and for all those features PCA parameters of 16, 32, and 64 dimensions. There is further optimization potential here. The audio features were calculated using the software AMUSE [66].

The linear SVM, the Random Forest with 100 trees, and the Naïve Bayes classifier were employed as basic classifiers. The models were validated based on the balanced classification error estimated during stratified cross-validation with k=5 partitions (see Section 3.2). The balanced error is estimated from applications of a classifier on a test data set, which is independent from the training data set, and is defined as
(5)$$\begin{array}{c} e_{bal}=\frac{1}{2}\cdot \left(\frac{c_{1,2}}{c_{1,1}+c_{1,2}}+\frac{c_{2,1}}{c_{2,1}+c_{2,2}}\right), \end{array}$$
where the parameters $c_{i,j}$, $i,j\in \left\{1,2\right\}$ are the entries of the *confusion matrix*, which summarizes the numbers of positive and negative predictions (Figure 2).

Stratified cross validation divides the available data set into k≥2 non-overlapping partitions [67]. In *k* runs, each partition, in turn, is used as test set and the other k−1 partitions form the training set, and the mean balanced test error across all runs is reported. Stratification ensures that the ratio of the different classes to predict in the partitions is approximately the same as in the given data set.

### 4.2. Visual Data Analysis

The results are presented as *heat maps* (Figure 3). The horizontal axis corresponds to genres and the vertical axis to feature combinations. The entries of the resulting matrix contain the balanced error rates, additionally visualized with colors. In each column, the minimum with the best configuration per genre is marked with a frame. The genres are sorted in ascending order by the minimum of the related column. The vertical axis is grouped into blocks of feature combinations of the same modality and combinations of several modalities. The blocks are sorted in ascending order based on the number of modalities; the first block contains only combinations of audio features, the second one—image features, the third one—text features, the fourth one—combinations of audio and image features, etc.

All results are visualized in the Section D.1. It is very difficult to provide general recommendations because of the large number of configurations and feature combinations. In order to reduce this effect, we propose three steps presented below: *aggregation of the same combinations of features*, *removal of dominated results*, and *filtering of less relevant results*.

#### 4.2.1. Aggregation of the Same Combinations of Features

To reduce the number of lines in the visualizations, the results are aggregated by combinations of features that use the same features but different configurations for them. For example, the combinations*SIFT_BOF (v = 400, pca = no) + TIMBRE*,*SIFT_BOF (v = 400, pca = 16) + TIMBRE*,*SIFT_BOF (v = 400, pca = 64) + TIMBRE*, 
aggregated as*SIFT_BOF + TIMBRE*,
correspond to a vector whose components are the minimal errors across all aggregated combinations of features for each classifier and each genre. This aggregation is in the following called *minimum accumulation*. Section D.2 shows the minimum accumulation for the individual classifiers.

#### 4.2.2. Removal of Dominated Results

It is desired to achieve the lowest possible error rates for each genre. The selection of features and their configurations can, therefore, be interpreted as a multi-objective minimization problem with *G* optimization criteria (errors for each genre). According to Zitzler et al. [68], a solution $K_{1}$ (feature configuration) *dominates* a solution $K_{2}$ if and only if the configuration $K_{1}$ has a better error rate $e_{K_{1}}$ than $e_{K_{2}}$ in at least one genre and no worse one in any other genre. Dominated feature configurations are not relevant for the investigation of certain hypotheses and can be removed from the views. The application of this method after minimum accumulation described in the previous section leads to Section D.3, Section D.4 and Section D.5. In Section D.6, the results of all classifiers have been compiled, then the same configurations of features have been aggregated, and, finally, the dominated configurations of features have been removed.

#### 4.2.3. Filtering of Less Relevant Results

As described in the previous section, the identification of the best feature groups can be understood as a multi-objective minimization problem with *G* objectives.

Let *r* be a *reference point* in the multi-objective space, which indicates the worst possible solution (all errors are equal to 1). When considering a solution *K* (selected feature group) in the objective function range, a volume exists with respect to *r* that is dominated by *K* (Figure 4a). All arbitrary solutions $K_{\mathrm{dom}}$ within this volume are dominated by *K*. This volume is called the “dominated hypervolume of solution *K*”. Likewise, a set K of solutions has a dominated hypervolume (Figure 4b). It is the volume, in which all objective function values of all arbitrary solutions are dominated by at least one solution in K.

Each non-dominated solution *K* from K contributes a part to the total dominated hypervolume of K, which is dominated exclusively by *K* (Figure 4c). This volume can be calculated. To every solution, i.e., every combination *K* of features, a share $v_{K}$ of the contribution to the total dominated hypervolume of K can, therefore, be assigned. A small $v_{K}$ is an indicator that there are further solutions near *K* in the objective function range.

Feature combinations with small $v_{K}$-values may be less interesting when examining the hypotheses, since there are other combinations of features whose classification error rates are similar to that of *K*. In order to further reduce the visualization of the test results, feature combinations *K* are removed, for which $v_{K}<t\cdot \max\left(v_{K^{\prime }}\;|\;K^{\prime }\in K\right)$, where t∈[0,1]. The application of this approach with t=0.01 and t=0.05 after the removal of non-dominated results leads to Section D.7, Section D.8, Section D.9, Section D.10, Section D.11, Section D.12, Section D.13 and Section D.14.

### 4.3. Feature-Related Hypotheses

The feature-related hypotheses are as follows:$M_{1}$:The classification with audio-based features achieves a better error rate than the classification with non-audio-based features. Feature combinations are not examined here.$M_{2}$:The combination of features of different modalities leads to a better error rate. More specifically:$M_{2,1}$:The combination of any features of two modalities results in a better error rate compared to using any features of one of the two modalities.$M_{2,2}$:The combination of any features of three modalities results in a better error rate compared to using any features of two of the three modalities.$M_{3}$:Non-audio-based features achieve a better error rate for certain genres whose error rate is high when classified via audio features.$M_{4}$:The use of principal component analysis for text and image features does not degrade the results with the respect to the classification error.

All hypotheses are examined via *Wilcoxon Signed Rank Tests* [69], checking whether the values of two paired samples are different. For this purpose, a *null hypothesis* $H_{0}$ and an *alternative hypothesis*$H_{1}$ are first set up. $H_{0}$ is an assertion about the observed error rates that the test is intended to refute. $H_{1}$ is the opposite of $H_{0}$, i.e., either $H_{0}$ or $H_{1}$ is true. The samples represent two observed error rates of different configurations corresponding to $H_{0}$. Then, a *significance level*
α, 0<α≤1, is chosen. It describes the probability of $H_{0}$ being incorrectly rejected by the test. Finally, the test is carried out. The result is a so-called *p*-value. If the *p*-value is below α, the test rejects $H_{0}$. The error rates examined differ significantly in this case. However, if the test does not reject $H_{0}$, this does not mean that $H_{0}$ is approved; rather, the null hypothesis is simply not rejected.

All hypotheses are examined with a commonly used significance level of α=5%. All tests are performed on the error rates of the individual classifiers to check whether some hypotheses can only be confirmed or rejected by using certain classifiers. Since all hypotheses are analyzed by multiple tests, the significance level for individual tests is further lowered by the Bonferroni correction, as described in reference [70] (p. 247).

Details of the procedure are described in the following analysis of hypothesis $M_{4}$. This is done before the analysis of the other hypotheses because the Bonferroni correction can be explained well on the basis of this hypothesis. The data basis for hypothesis $M_{4}$ is the error rates shown in Section D.1. The null hypothesis $H_{0}$ of the test is that the error rate remains the same when using PCA. To test $H_{0}$, sub-hypotheses are set up comparing configurations with and without PCA. Examples are:$H_{0,1}$:The use of BoW features without PCA achieves the same error rate as the use of BoW features with a PCA with dimensionality reduction to 64 dimensions.$H_{0,2}$:The use of BoW features without PCA achieves the same error rate as the use of BoW features with a PCA with dimensionality reduction to 32 dimensions.$H_{0,3}$:The use of BoF features without PCA achieves the same error rate as the use of BoF features with a PCA with dimensionality reduction to 64 dimensions.

$H_{0}$ must be rejected as soon as at least one of $H_{0,1},H_{0,2},\ldots ,H_{0,k}$ is rejected. Let $\alpha_{k}$ be the level of significance, with which the tests on the hypotheses $H_{0,1},H_{0,2},\ldots ,H_{0,k}$ are performed. Then, there is the likelihood of falsely rejecting one of these hypotheses at $1-{\left(1-\alpha_{k}\right)}^{k}$. If we want to test α on $H_{0}$ with a significance level α, then, $\alpha_{k}=\alpha /k$ can be chosen because $1-{\left(1-\alpha /k\right)}^{k}<\alpha$ for k>1. The Bonferroni correction describes this procedure. Instead of changing the significance level $\alpha_{k}=\alpha /k$, the *p*-value obtained by the test can be equivalently also adapted to $p_{k}=p\cdot k$. The Bonferroni correction is used in all subsequent tests.

To test *hypothesis $M_{4}$*, the error rates are tested against each other using different feature configurations $K_{1}$ and $K_{2}$. $K_{1}$ will be tested against $K_{2}$ if all of the following conditions are true:$K_{1}$ and $K_{2}$ are no combinations of individual features groups.$K_{1}$ and $K_{2}$ are only features of type BoW, doc2vec, or BoF.$K_{1}$ and $K_{2}$ are the same feature type.$K_{1}$ does not use PCA, $K_{2}$ uses PCA.

The results of the tests can be found in Table A10, Table A11 and Table A12. For all tests, the null hypothesis is retained for all classifiers. Thus, the results using PCA do not differ significantly from results that did not use PCA. Hypothesis $M_{4}$ is, therefore, not rejected, meaning that the number of features can be significantly reduced without a decrease of the classification performance.

For the analysis of *hypothesis $M_{1}$*, only those feature configurations are considered from Section D.2, which do not consist of feature combinations. These are then partitioned by modality. The individual partitions are summarized by minimum accumulation (see Section 4.2.1). This results in three vectors $e_{a}$, $e_{i}$, $e_{t}$ of error rates for audio, image, and text features. Then, using the Bonferroni correction $e_{a}$ against $e_{i}$ and $e_{a}$ against $e_{t}$ are tested. The results of the tests can be found in Table A1, Table A2 and Table A3.

For Random Forest and the Naïve Bayes classifier, at least one of the null hypotheses is rejected. Audio features in this case provide results that are significantly different to text or image features. Since, in any case, the median error rate is lower when using audio features, we can agree with hypothesis $M_{1}$, at least when using Random Forest or Naïve Bayes classifier.

*Hypothesis $M_{2,1}$* is checked by partitioning all error rates of the feature configurations from Section D.2 to modality combination. This results in the partitions *audio*, *text*, *image*, *audio* + *text*, etc., which are combined by minimum accumulation to form error rate vectors $e_{a}$, $e_{t}$, $e_{i}$, $e_{\mathrm{at}}$, etc. Now, all error rates $e_{m}$ are tested against $e_{n}$, for which

$e_{m}$ belongs to a partition of exactly one modality (e.g., *audio*),$e_{n}$ belongs to a partition of two modalities (e.g., *audio* + *text*),the modalities of the partition of $e_{n}$ include the modality of the partition of $e_{m}$.

The results of the tests adjusted by the Bonferroni correction can be found in Table A4, Table A5 and Table A6. For all classifiers, at least two of the six resulting null hypotheses are rejected. Thus, there are partially significant differences in the error rates when using features of different modalities compared to the error rates when using features that belong to only one modality. For each rejected null hypothesis, the error rates using features of two modalities show a lower median, so the error is significantly better here. Hypothesis $M_{2,1}$ cannot always be approved, as not all null hypotheses are rejected. However, if we restrict ourselves to certain classifiers and modalities, such as Random Forest with audio and text features, the hypothesis can be approved. Accordingly, it seems to apply only in certain scenarios.

Considering the tests individually without using the Bonferroni correction, it is worth noting that almost every null hypothesis is rejected in favor of the combination of features of different modalities. Exceptions are the null hypotheses of the tests, which test the use of text features against the use of text and image features and the use of audio features against the use of audio and text features in classification via an SVM. It is also noticeable that the median error rate is 5% to 17% lower compared to using a non-audio feature when the non-audio feature is combined with an audio feature. Overall, it is apparent that the combination of features of two modalities almost invariably leads to an improvement in the error rate, whereby the inclusion of audio features in the feature combination seems to lead to the greatest improvement in error.

*Hypothesis $M_{2,2}$* is investigated analogously to hypothesis $M_{2,1}$. Partitions by modality combination are created and again summarized by minimum accumulation. All error rates $e_{m}$ against $e_{n}$ are tested, for which

$e_{m}$ belongs to a partition of exactly two modalities (e.g., *audio* + *text*),$e_{n}$ belongs to a partition of exactly three modalities (e.g., *audio* + *text* + *image*),the modalities of $e_{n}$’s partition include all modalities of the partition of $e_{m}$.

The results of the tests adjusted by the Bonferroni correction can be found in Table A7, Table A8 and Table A9. For all tests, the null hypothesis is retained for all classifiers. Neglecting the Bonferroni correction, it turns out that taking image or audio features into the feature combination with SVM as a classifier and including text or audio features in the feature combination using Naïve Bayes shows a statistically significant improvement in the classification quality. Altogether, contrary to the observations of hypothesis $M_{2,1}$, hypothesis $M_{2,2}$, therefore, cannot be generally confirmed. Nevertheless, the combination of features of three modalities in the cases mentioned brings an improvement in the error rate. For hypothesis $M_{2}$ as a whole, the combination of features of two modalities certainly brings an improvement in the error rate. However, adding more modalities does not necessarily improve the classification performance significantly.

For *hypothesis $M_{3}$*, from Section D.2 only error rates of the genres R&B, Reggae, Pop, and Electronic are considered. These genres were chosen because none of the classifiers are able to achieve error rates below 25% using only audio features. Since there are only four observations per potential test, no tests can be used. For this reason, this hypothesis is assessed using Section D.2. Considering the error rates of the classifiers using image or audio features only for the selected genres, it is easy to see that the use of non-audio-based features does not effect any noticeable improvement. Although some combinations of image features may bring an improvement to the reggae genre, this seems to be an exception, so that hypothesis $M_{3}$ is generally unconfirmed.

### 4.4. Classifier-Related Hypotheses

The classifier-related hypotheses are as follows:$M_{5}$:The different classification methods have different error rates for the same features.$M_{6}$:There are genres, for which certain classifiers achieve a better error rate for the same features than other classifiers.

To study *hypothesis $M_{5}$*, we first consider Section D.2. Here, the classification performances of the three classifiers with all feature combinations are shown. A first visual impression conveyed by the color coding is that the Naïve Bayes classifier delivers results that differ significantly from the results of the other classifiers. On a majority of genres, the classification error appears to be higher than the error of SVM and Random Forest. This is also evident from the absence of the yellow-orange block in the left-hand part of the diagram, which arises in the charts of SVM and Random Forest in that certain genres can be better classified almost independently of the feature selection. Looking at Section D.6, which summarizes the data in Section D.2 and outlined feature combinations, this assumption is confirmed. Most of the results of the Naïve Bayes classifier are dominated by other results. In Section D.14, feature combinations are removed that are less than t=0.05 contributing to the dominated hypervolume of the total set. In this figure, no result of the Naïve Bayes classifier is listed. Therefore, on the features studied here, this classifier generally appears to provide higher error rates compared to SVM and Random Forest, so hypothesis $M_{5}$ can be agreed.

*Hypothesis $M_{6}$* is checked using Section D.2. The sorting of the genres on the horizontal axis of the three visualizations is different, so the classifiers have different best error rates per genre. It is striking that the genres Rap/Hip-Hop, Classical, and Jazz are among the three genres of all classifiers that can be classified with the lowest error rate. The genres Pop, R&B, and Electronic are among the genres with the highest classification error rate for all classifiers. Thus, there seem to be tendencies of classification quality per genre, which are independent of the used classifier. However, there are also strong differences in the error rates of the individual classifiers. Random Forest provides noticeably better error rates on the Rock genre than the Naïve Bayes classifier and SVM when using feature combinations that include audio features.

The Naïve Bayes classifier, on the other hand, tends to achieve an error rate of approximately 50% in many genres when feature combinations with audio features are used. Further investigations show that this error rate arises because the classifier always classifies tested music pieces as not belonging to the genre to predict. This may be explained by the fact that audio features may have correlations with each other, which the Naïve Bayes classifier cannot handle. It seems, therefore, that the Naïve Bayes classifier with audio features is a non-recommendable configuration for a genre recognition system. Overall, hypothesis $M_{6}$ can be approved.

## 5. Conclusions and Future Work

We have proposed a multi-modal genre recognition framework that considers the modalities audio, text, and image by features extracted from audio signals, album cover images, and lyrics of music tracks. The basis of recognition is binary classification, and the well-known and proven classifier methods (Naïve Bayes, Support Vector Machine, and Random Forest) were chosen for this purpose. Features were selected that are known to be particularly powerful in the domains of audio signal, text, and image, and an approach to their combination that meets the requirements of the features of the different modalities was presented.

Extensive experiments have been conducted for the three classifier models and numerous feature combinations. As no suitable data collection was available, an in-house multi-modal data set was compiled from several music collections. Determining the feature values required some effort, but it should be noted that the feature values are reusable. On the other hand, the training and application of the classifiers required comparatively little time. The training runtimes for the three classifiers used are low, compared to those often observed for end-to-end classifiers, such as deep neural networks.

The influence of the classifiers and the features on the classification quality was assessed by using the balanced classification error. The error values were presented visually by tables with color coding. Three approaches to data reduction were applied: aggregation of the combinations of the same, but differently configured features, removal of dominated results based on multi-objective non-dominated sorting of selected combinations of features and classifiers, and removal of less relevant results with small hypervolume contributions. The approach has proven successful for comparative visual analysis by allowing the range from a heatmap-like overview based on the original data to a detailed table-based view based on the reduced data.

The statistical comparison of all combinations of two modalities against individual ones always led to smaller classification errors. Those errors were also significantly smaller for all cases, except for text and audio modality against audio, and text and image against text using SVM. A more general hypothesis that “two modalities are always better than one” was confirmed by adjusted *p*-values after the Bonferroni correction for multiple tests for half of all combinations. The extension to the third modality further reduced the errors in almost all cases, but the general hypothesis that “three modalities are always better than two” could not be confirmed by adjusted *p*-values; the advantage rather depends on the classifier and features used.

For more robust genre recognition and music recommendation systems, future work should further extend the number of modalities (e.g., integrating MIDI scores, music videos, meta data), feature groups, and classification methods. To better understand the characteristics of music categories, it is possible to build and compare distinct feature sub-groups based on musical and statistical properties, extraction costs, availability in open-source frameworks, etc. Deep features can be extracted not only from the last hidden layer of the previously trained network but also from other layers, as proposed by Choi et al. [71]. For a more efficient identification of the best classification models, feature selection and systematic tuning of classifiers can be further applied. In addition, the experiments can be repeated using further data sets and genres or also other music categories, such as emotions or personal preferences. Last but not least, the demands on resources (runtime, storage space) can be measured.

## Figures and Tables

**Figure 1 entropy-23-01502-f001:**
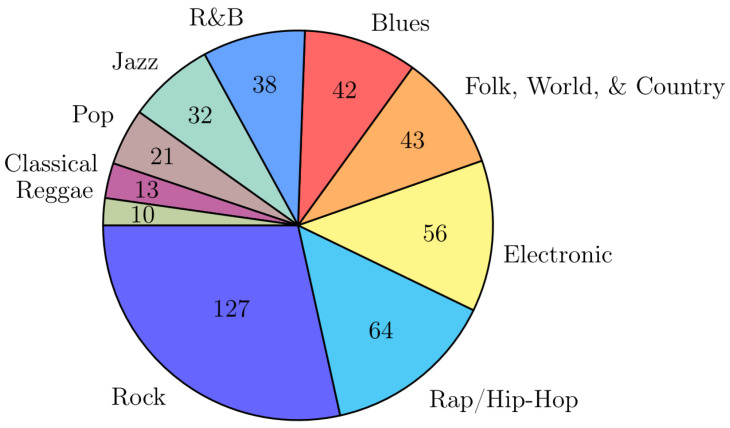
Composition of the genres of the created data set.

**Figure 2 entropy-23-01502-f002:**
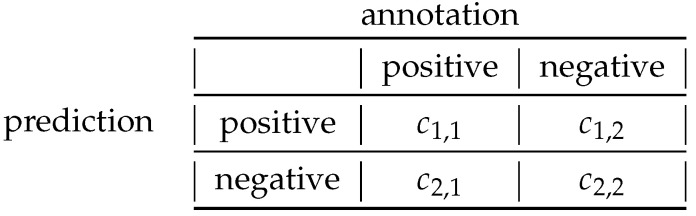
Confusion matrix of a binary classification problem.

**Figure 3 entropy-23-01502-f003:**
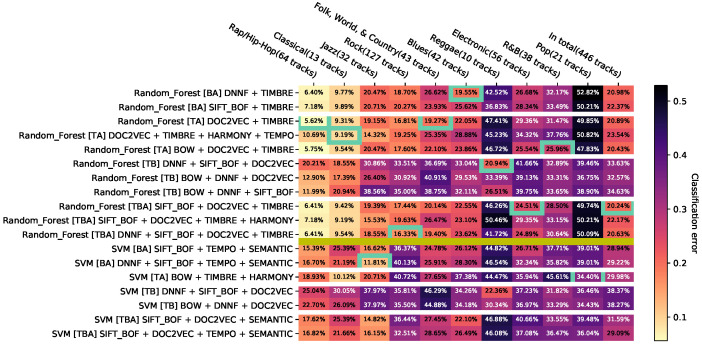
Visualization example.

**Figure 4 entropy-23-01502-f004:**
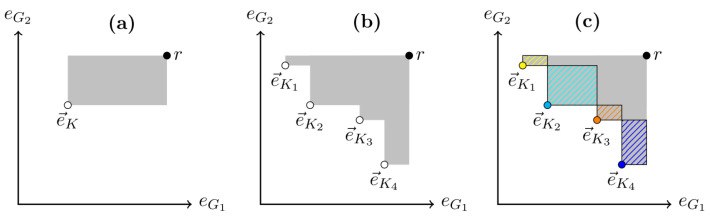
Visualizations of dominated hypervolumes with respect to the reference point *r* with a two-objective minimization problem. (**a**) Dominated hypervolume of a solution. (**b**) Dominated hypervolume of a set of solutions. (**c**) Individual contributions of solutions to the dominated hypervolume.

**Table 1 entropy-23-01502-t001:** Configurations of BoW-features.

	Vocabulary Line Size	Stop Word Line Removal	Stemming	TF-IDF	PCA
Configuration 1	400	yes	yes	yes	no
Configuration 2	400	yes	yes	yes	32

**Table 2 entropy-23-01502-t002:** Configurations of doc2vec-features.

	Size of the Hidden Layer	PCA
Configuration 1	100	no
Configuration 2	100	16

**Table 3 entropy-23-01502-t003:** Configurations of the BoF-features with SIFT-descriptors (SIFT_BOF).

	Vocabulary Size	PCA
Configuration 1	400	no
Configuration 2	400	16
Configuration 3	400	64

**Table 4 entropy-23-01502-t004:** Configurations of the features from deep neural networks (DNNF).

	PCA
Configuration 1	32
Configuration 2	64

## Data Availability

In case the article will be accepted, we will release the data set with all extracted features.

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
