# Peer review of "Statistical and Visual Analysis of Audio, Text, and Image Features for Multi-Modal Music Genre Recognition"

_entropy, 2021, doi:10.3390/e23111502_

Round 1

Reviewer 1 Report

The paper is interesting as the combination of audio features, album images, and lyrics is still rarely seen in the literature. It also fits the journal scope of interest.

Title: Statistical and Visual Analysis of Audio, Text, and Image

Features for - I am not sure whether the tile focuses on the main goal of the study; maybe it would be better to change its order   “Multi-Modal Music Genre Recognition based Audio, Text, and Image

Features”?

After such a thorough analysis in Section 4 that covers a lot of material, Conclusions and Future Work are somewhat disappointing. I suggest including some general comments concerning the analyses performed. Moreover, “future work” is written in a style that does not communicate any useful information. I propose to rewrite the last paragraph of this Section and include more details. You may refer to problems and difficulties that should be further analyzed.

Did you evaluate how time-effective your method is? This issue should be addressed.

General comments:

The first (and to some extend 2nd) paragraph in Introduction seems obsolete. I suggest starting with the third one and then passing on the Related work.

The same remark concerns Section 3.2 – it covers well-known issues. At least, it may be shortened.

Although the list of References is quite long, the state-of-the-art is not complete (see comment below).

There are quite many language issues in the text. I have listed some of them beneath. Moreover, notably, punctuation and article should also be checked.

I propose to use References instead of Footsteps.

Starting a sentence with square brackets containing reference number is not very comprehensible. It is better to include the author’s name, i.e., “According to Sturm [1], ….

I am not sure about Appendices; maybe they should be placed under the Internet link?

At least, References should be before Appendices.

Detailed remarks:

Abstract:

 mostly the audio -> mainly

to a pure learning -> to pure learning

a higher interpretability -> higher

and a futher -> and further

increase of -> in

Section 1

As features to represent music tracks, usually audio signal characteristics are calculated. -> not comprehensible

“independently of its popularity”??? -  is this a necessary statement?

Section 2

“The use of album covers to determine music genres has been less investigated until  now…” and “However, none of these approaches took albums covers into account.” -> There are a few recent literature sources that analyze relationship between album covers and music genres, they should be contained in References

we take also -> we also take

less parameters -> fewer

only few -> only a few

is is

meta data -> meta data

15 second -> 15-second

This sentence is not comprehensible: “Second, we estimate text features for lyrics and not album reviews; although in [2] it is argued that relevant genre information must not be captured in reviews and “will unlikely comply with the current taxonomy of the collection to be classified”, is is indeed probable that some reviews may contain such information and be compiled from many other mixed sources (experiences of a reviewer, meta data, artist information, etc.), so that lyrics can be rated as a more “pure” and “independent” modality.”

Section 3

a rigorous -> rigorous

by certain -by specific

can be always automatically -> can always be automatically

the Figure -> Figure

Figs. 2 and 3 – I am not sure whether “item” is an appropriate word here

A better measurement of classification quality -> evaluation?

cross validation -> cross-validation

A prominent example of such models are -> is

in phase domain -> in the phase domain             

predicted also -> also predicted

the high comprehensibility -> high comprehensibility

possibly -> possible

a training  set -> the

the final  prediction  is done -> made

Section 4

of the classifier types??? Naïve Bayes, SVM, and Random Forest -> types, i.e., …?

in the Tables 1 and 2 -> in Tables 1 and 2

features BoW -> BoW features

in the Tables 3 and 4 -> in Tables 3 and 4

of the image features BoF SIFT -> of the image BoF SIFT features

the second one image features, … -> the second one − image features,  etc.

worth to note -> worth noting

any obvious -> any noticeable

Reviewer 2 Report

The authors introduce a fusion technique to predict genres based on audio tracks, album cover images, and lyrics. 
In the experimental section, the combination of the three modalities showed good performance in several cases.

Both content organization and experimentation are evaluated as well-organized.

Round 2

Reviewer 1 Report

Thank you for taking into account my comments. I appreciate that. In my opinion, the text is now much better organized and its aims are clearly presented.